# Effect of CNTs Additives on the Energy Balance of Carbon/Epoxy Nanocomposites during Dynamic Compression Test

**DOI:** 10.3390/polym12010194

**Published:** 2020-01-11

**Authors:** Manel Chihi, Mostapha Tarfaoui, Chokri Bouraoui, Ahmed El Moumen

**Affiliations:** 1ENSTA Bretagne, IRDL (UMR CNRS 6027) / PTR-1, F-29200 Brest, France; ahmed.el_moumen@ensta-bretagne.fr; 2LMS, ENISo, University of Sousse, Sousse 4023, Tunisia; chokri.bouraoui@enim.rnu.tn

**Keywords:** CFRP, Carbon nanotubes, Nanocomposites, Split Hopkinson Pressure Bar, Energy absorption

## Abstract

Previous research has shown that nanocomposites show not only enhancements in mechanical properties (stiffness, fracture toughness) but also possess remarkable energy absorption characteristics. However, the potential of carbon nanotubes (CNTs) as nanofiller in reinforced epoxy composites like glass fiber-reinforced polymers (GFRP) or carbon fiber-reinforced polymers (CFRP) under dynamic testing is still underdeveloped. The goal of this study is to investigate the effect of integrating nanofillers such as CNTs into the epoxy matrix of carbon fiber reinforced polymer composites (CFRP) on their dynamic energy absorption potential under impact. An out-of-plane compressive test at high strain rates was performed using a Split Hopkinson Pressure Bar (SHPB), and the results were analyzed to study the effect of changing the concentration of CNTs on the energy absorption properties of the nanocomposites. A strong correlation between strain rates and CNT mass fractions was found out, showing that an increase in percentage of CNTs could enhance the dynamic properties and energy absorption capabilities of fiber-reinforced composites.

## 1. Introduction

Energy absorption is considered to be one of the most important functions of structural materials, especially when subjected to an accidental collision or sudden shock. It is a crucial condition for structural crashworthiness and damage assessment for example, in designing rail cars, aircraft, automobiles and rotorcraft. During the design phase, the crashworthy structure is manufactured in such a manner that it can halt the transfer of the energy to the passenger compartment by absorbing all the impact energy in a controlled manner during crash. Moreover, in civil construction, many structures lack energy-absorbing capabilities, resulting in catastrophic failure during events like explosions. This can cause massive human causalities and property loss. However, this can be avoided by improving the blast resistance of buildings, using sacrificial cladding structures with cores made of highly shock-absorbing material.

Traditionally, structural components used in crashworthy applications and armors were commonly manufactured using metals, because metals are able to absorb the impact energy in a controlled way because of their high toughness [1]. Then, researchers started using cellular forms such as honeycomb structures, foams and sandwich structures, which have demonstrated excellent resistance to dynamic loading due to their bulking and collapse mechanisms [2,3,4,5,6]. Furthermore, composite structures have shown excellent resistance against vibrations during impact thanks to their unique internal structure, which displays good internal damping behaviors [7,8]. These composites have been further enhanced by the introduction of nanotechnology and nanofillers. The introduction of nanoparticles into polymer matrices improved their thermal and electrical properties as well as enhancing their mechanical characteristics, such as strength and toughness. In addition, recent research showed that reinforcing the polymeric materials with nano-fillers such as CNTs resulted in enhancement of structural damping of the composites because of the large surface-to volume ratio of nanofillers which can result in exceptional performance of interfacial bond between the nanofillers and matrix resulting in an increase of the energy dissipation capability of the material [9,10,11,12,13]. Furthermore, members of the fullerenes family exhibit extraordinary energy absorption behavior because of their high strength, stiffness, and large surface area [14,15,16,17,18,19,20]. This is why nanocomposites with distinct matrices and filler materials show improvement not only in stiffness and fracture toughness during experimental studies, but also in impact energy absorption and vibration damping. For this reason, these nanocomposites have significant importance in civil and military applications like automobile, airplane structures and biomedical [11].

Among these nano-sized inclusions, carbon nanotubes (CNTs), being members of the carbon nanomaterials family, have shown unique energy absorption performance when used in 3D sponge-array and foams architectures because of their unique mechanical properties [20,21,22,23]. Additionally, CNTs have an ultra-high stiffness, strength and an extremely large surface area. They are also extremely lightweight in comparison to traditional materials. In fact, both experimental and computational results have shown that they had about tensile strength of 200 GPa, Young’s modulus of 1 TPa, shear modulus of 1 GPa, bulk modulus about 462–546 GPa and bending strength approximately 14.2 GPa [15,16,24]. Recent studies also proved that nanocomposite laminates based on nanocharge materials have good overall energy absorption characteristics. Numerous previous experimental works have shown that nano-fillers are able to enhance not only stiffness but also the energy absorption behavior of polymers and/or conventional composites [25].

Drdlova and Prachař [26] studied the mechanical performance of lightweight porous foams reinforced with carbon nanotubes with 1–5 vol.% for structural applications under high strain rate loading using SHPB and results had shown that energy absorption capability of the material was greatly enhanced up to 4 vol.%, and then there was a significant decrease.

Chen et al. [27] developed a numerical model using dynamic simulation to study the energy absorption ability of CNT bucky paper under high-velocity impacts. Their study revealed that this bucky paper showed extremely high kinetic energy dissipation efficiency within its elastic limit, and that this depended directly on the impact velocity. In addition, Chen et al. [28] also studied the energy dissipation behavior of CNTs with nested bucky balls during impact using a dynamic simulation model. The simulated results showed that dissipated energy was mostly converted into the thermal energy at low velocity impact while bucky balls showed permanent strain deformation at high velocity impact; thus, dissipation energy was dominated by the strain energy of the energy absorption system.

Weidt et al. [29] performed a study using 2D and 3D computational modelling on aligned CNT/epoxy nanocomposites under compressive strain rates. The results revealed that, by increasing the wt.% of CNTs, the nanocomposites showed a noticeable increase in their mechanical performances including energy absorption behavior.

Although considerable research has been devoted to the energy absorption behavior of CNT-based nanocomposites, rather less attention has been paid to the dynamic/impact properties. As far as the compressive response of the composite is concerned, the Split Hopkinson Pressure Bar (SHPB) technique is one of the examples and has been extensively used to evaluate the impact behavior of different materials at high strain rates [30,31,32,33,34,35,36,37,38,39,40]. For instance, Gardea et al. [41] showed in their investigation that strain energy dissipation of CNT reinforced polymers under low strain was not dependent on the alignment of CNTs; however, damping factor increased monotonically with the wt.%. of CNTs, which showed the occurrence of friction dissipation mechanisms within the CNT–CNT interface; however, interfacial slip contributed to energy dissipation at higher strain rates. Moreover, they showed that the tearing and plasticity of the matrix caused by the misaligned CNTs within the loading direction played a vital role in energy dissipation. Gardea et al. [42] in another study showed that carbon nanotube (CNT)-reinforced acrylonitrile-butadiene-styrene (ABS) composites fabricated by additive manufacturing exhibited of strain energy dissipation ability with reduced damage because of the CNTs. Their results showed that CNTs altered the energy dissipation mechanism and controlled the structural damping behavior under dynamic loading. El Moumen et al. [43] evaluated the effect of integrating the CNTs in epoxy on its shock wave absorption under dynamic compressive loading using SHPB. The results showed that as the wt.% of CNTs increased, the nanocomposites were able to absorb more mechanical shock waves. Thus, this highlights the importance of CNTs in enhancing the impact resistance behavior of the composite structures. It was also found in a study that the aspect ratio and mass fraction of CNTs played a vital role in defining the energy absorption characteristics of a nanocomposite under high strain rate impacts [29].

However, the potential of CNTs, as nanofiller in reinforced epoxy composites like glass fiber-reinforced polymers (GFRP) or carbon fiber-reinforced polymers (CFRP) under dynamic testing, is still underdeveloped. There is very little, if any, information available in the literature. Therefore, in this context, the object of this paper is to study the effect of using various weight percentages of CNTs in CFRP composites on their shock wave or energy absorption performances. An experimental study was performed to investigate the energy absorption behavior of CNTs-based CFRP nanocomposites under out-of-plane dynamic loading, using the SHPB device. Samples were fabricated with 1 and 2% mass fractions and specimen with 0% was considered as a reference. Moreover, these dynamic compression tests were executed at three different impact pressures, i.e., 2, 3 and 4 bar, to further analyze the energy absorption ability of these nanocomposites.

## 2. Materials and Manufacturing Process

The polymer used in this study was a low-viscosity liquid epoxy resin, Epon 862 (Diglycidyl Ether of Bisphenol F), acquired from Momentive Specialty Chemicals Inc. (Cleveland, OH, USA). The carbon fiber was provided by Hexcel Company and multi-walled carbon nanotubes (MWNTCs) were produced by Nanocyl Belgium Company (Sambreville, Belgium), they were synthesized with no surface functionalization; they had an average diameter of 10 nm and length of 1.5 μm. Mechanical properties of each constituent are listed in Table 1.

Figure 1 shows the SEM and TEM (University of Dayton, United States) characterization of CNTs in epoxy resin at micro and nano scales. The multiwall nanotubes were tube-shaped materials and considered as long curved cylindrical fibers (snake-like shapes). The CNTs are randomly distributed into matrix, Figure 1a. Transmission electron microscopy (TEM) of CNTs shows the fiber shape, see Figure 1b.

The fabrication of the nanocomposites consisted of, first, dispersing CNTs in the polymer matrix, varying the weight fraction of MWNTCs between 0 and 2%, and then mixing this material using an T25 digital ULTRA-TURRAX increased shear laboratory mixer for a total of 30 min at 2000 rpm. Afterwards, an ultrasonic bath was also used, and the mixed material was further processed in a Lehmann Mills three-roll mixer (University of Dayton, United States) to guarantee a homogeneous dispersion of CNTs (Figure 2), the film with 120 µm in thickness containing CNTs was manufactured using film line, Figure 3a. The reinforced epoxy was introduced with the 5 HS (satin) T300 6k carbon fiber fabric, using the infusion process; Figure 3b,c. The reinforced epoxy resin flowed between the fiber plies, and the press curing condition was set to 200 MPa. All panels manufactured consisted of 24 carbon fiber fabric layers interleaved with 25 layers of CNTs/epoxy film to accomplish an overall fiber volume fraction of 50%. The panels were then cooled. SEM characterization was performed to demonstrate the CNT distribution with 500 nm resolution. The SEM image confirms the random distribution of CNTs with variable length; Figure 3d,e.

Samples with dimensions of 13 mm × 13 mm × 8 mm were then cut from the prepared specimen plates for out-of-plane compression test on SHPB, Figure 4.

## 3. Test Procedure

### 3.1. SHPB Test Method

Figure 5 is a schematic of The Split Hopkinson Pressure Bar (SHPB) apparatus that was used in this study to assess the shock wave absorption characteristics of the specimens. The experimental setup was composed of striker, incident (input) and transmitted (output) steel bars. A compressive longitudinal wave was induced in the incident pressure bar by impacting it with the striker bar at a specified velocity (Impact energy). A compressive incident wave $\varepsilon_{I}\left(t\right)$ was generated when a striker bar impacted the free end of the input bar and travelled across the input bar until it got to the bar-specimen interface. Once the specimen was hit by the incident wave, the wave was split into two parts. One part was transferred to the output bar as a compressive transmitted wave $\varepsilon_{T}\left(t\right)$ and the other part was reflected to the input bar as a tensile reflected wave $\varepsilon_{R}\left(t\right)$. These three pulses $\varepsilon_{I}\left(t\right)$, $\varepsilon_{R}\left(t\right)$ and $\varepsilon_{T}\left(t\right)$ were measured using strain gauges mounted at the middle of each pressure bar, and a digital oscilloscope was used for data acquisition; Figure 6. Recorded data were then treated using by means of the Maple Software algorithm to acquire all dynamic parameters like, for example, forces and velocities, as functions of time at the two faces of the specimen, which had already been sandwiched between the two pressure bars.

### 3.2. Dynamic Compression Testing

During the experimental investigation, the velocity of the striker bar (impact pressure) was controlled to obtain a wide range of impact energy magnitudes. Variation of the incident velocity as a function of time was plotted to assess the dynamic response of each specimen at impact pressure P = 4 bar, and the results confirmed the test reproducibility, which was common for all CNTs weight percentages; Figure 7. We performed tests at different pressures ranging from 2 to 4 bar but 4 bar was the pressure at which damage was exhibited in the samples. We wanted to see the effect of CNTs on the improvement of the damage mechanism of fiber-reinforced composites, so 4 bar was chosen to present the diversity of the behavior of our material with respect to the addition of CNTs.

### 3.3. Theorical Characterization of Absorbed Energy

During the dynamic compressive tests, the input energy (impact energy) corresponding to the transferred kinetic energy of the striker bar to incident bar was obtained by varying its initial velocity. Once the input energy reached the bar specimen interface, it was split into two parts. One part was absorbed by the sample and could have caused plastic deformation or damage in different forms, which in turn could have led to heat generation if macroscopic damage occurred. The other part was transferred to the output bar as the transmitted energy [44]. Absorbed energy was the difference between the work transferred to the specimen from the incident bar and the mechanical work done by the specimen and transmitted to output bar [45]. The mechanical powers at the interfaces of two pressure bars were obtained by multiplying the corresponding velocity by the contact force.

Velocity (*V*) was determined using the incident and transmitted strains ($\varepsilon_{I}$ and $\varepsilon_{T}$) as stated by Park et al. [45]:(1)$$V\left(x,t\right)=c\;[-\varepsilon_{i}\left(t-\frac{x}{c}\right)+\varepsilon_{r}\left(t+\frac{x}{c}\right)]$$
where $c={\left(E/\rho \right)}^{\frac{1}{2}}$ is the longitudinal wave velocity of the bar, *E* is the Young’s modulus, and ρ is the mass density of the pressure bar.

The normal force (*F*) on any cross-section *x* of the incident and transmitted bars is:(2)F(x,t)=AEε(x,t)
where *A* represents the cross-sectional area of the bar.

It was obvious that the physical properties could be obtained only if the values of $\varepsilon_{i}$ and $\varepsilon_{r}$ were known. Hence, the main purpose of this experimentation was to find out these functions using strain gauges, mounted at the middle of each pressure bar. A digital oscilloscope was also used for data acquisition.

The overall mechanical work was calculated by integrating the mechanical power with respect to time. Thus, both the work transferred from the input bar to the CNTs-based nanocomposites sample ($W_{inc}$) and the work done by the sample and transferred to the output bar ($W_{trans})$ are given by [45]:(3)$$W_{inc}=-\int_{t_{i}}^{t_{f}}F_{inc}\left(t\right)V_{inc}\left(t\right)dt$$
(4)$$W_{trans}=\int_{t_{i}}^{t_{f}}F_{trans}\left(t\right)V_{trans}\left(t\right)dt$$
where the incident and transmitted physical parameters are presented by the subscript “*inc*” and “*trans*”.

The absorbed energy of the sample $\left(E_{abs}\right)$ was calculated using the equation below, given in [45]:(5)$$E_{abs}=W_{inc}\left(t\right)-W_{trans}\left(t\right)$$

Figure 8 shows an example of a typical absorbed energy curve for the test conducted on 0% CNTs sample. Results showed an increase in the absorbed energy during the impact and this energy absorbed by the material was the combination of two different energies. The first part was the elastic part, which was released until it reached a constant value (gradual unloading cycle). The second part was the inelastic unrecoverable energy, represented by the constant value, that was dissipated permanently through damage (the end of the cycle). Thus, the absorbed energy can be given as:(6)$$E_{abs}=E_{elas}+E_{diss}$$

## 4. Results and Discussion

A wide variety of input energies (or incident energy) was used to impact the carbon/epoxy nanocomposite specimens in order to understand the effect of CNTs on their energy absorption capability. Therefore, the energy absorption behavior of each specimen was studied at impact energy of 15 J, 34 J and 53 J. The absorbed energy results showed that all the samples had similar characteristics, Figure 9. This response indicated that no macroscopic damage was occurred as shown by Tarfaoui et al. [46]. Moreover, the fluctuation in the curves was caused by the storage and release of strain energy during the out-of-plane compression tests. Another interesting phenomenon observed is that the absorbed energy was very much influenced by increasing the impact energy, and the maximum peak also increased as the input energy was increasing. This behavior was common for all mass fractions of CNTs.

It can be seen that there was a noticeable effect due to the introduction of CNTs. An important portion of the impact energy $W_{imp}$ was absorbed during the impact. Actually, this absorbed energy was the energy stored in the specimen during the elastic deformation and it was released in the form of recoverable elastic strain energy and dissipated energy. Moreover, the experimental behavior of the CNT-based nanocomposites also confirmed that an increase in absorbed energy was observed because of the increase in both elastic and dissipated energies. However, the greatest portion of the absorbed energy was stored and released as elastic strain energy $(E_{elas}$) and only a small portion was dissipated energy $\left(E_{diss}\right)$. This response indicated that neat epoxy showed more plastic deformation instead of elastic behavior, compared to samples with a different mass fraction of CNTs. Addition of CNTs improved the elastic behavior of composites and reduced their plastic deformation as it became more resilient. In fiber reinforced composites the matrix is responsible for the plasticity because of their ductile nature. However, when CNTs were added as nanofillers in the matrix epoxy the material became more rigid with an increase in its elastic properties and reduction in its plasticity. In addition, CNT-reinforced nanocomposites had higher energy dissipation performance because of the increase in micro cracks. The CNTs behave as barrier for any crack propagation. Thus, they stop the propagation of any crack initiation, which can result in significant micro cracks instead of fatal macro damage and could delay the final fracture. This showed that, even with small weight percentage such as 1%, CNTs could improve the energy absorption of the CFRP laminate composites and delay the final fracture. Figure 10 gives a summary of the obtained results.

The energy dissipation caused in a nanocomposite during out-of-plane compressive loading was calculated by taking the area under the curve of the absorbed energy profiles for each specimen. Results revealed that there was an increase in dissipated energy as the wt.% of CNTs increased. The plausible scenario for the augmentation of the energy dissipation was the sliding phenomenon at the interface between CNTs and polymer matrix. Low mass density of CNTs and exceptionally large contact area at the interface between CNTs and matrix caused the frictional sliding at the CNT-CNT and CNT-matrix interfaces which could be the main cause of the increase in energy dissipation with minimal weight penalty [47]. Moreover, this sliding between the matrix and the CNTs could enhance the structural damping of the material. Recent studies of polymer material reinforced with nanofillers have also demonstrated that integrating nanofillers in the polymer matrix increased the damping of composite structures more efficiently [48].

Another method for augmenting the frictional energy dissipation is by boosting the weight fraction of CNTs in the composite; Figure 11.

## 5. Failure Mode

The experimental investigation was studied in detail to improve the understanding of the dynamic behavior of these nanocomposites during out-of-plane compression tests. The results showed that the dynamic properties of CNTs reinforced nanocomposite were significantly affected by increasing CNTs mass fraction. Strain rate and stress as functions of time were superposed for each mass fraction of CNTs at impact pressure of 4 bar, Figure 12. According to this figure, we are able to differentiate a variety of zones for each specimen and each zone is described individually as follow:

Zone 1: strain rate increases quickly before attaining the highest peak. This maximum value was reduced through the increase of CNTs mass fraction and this can be explained by the increase in the rigidity of the matrix material because of the presence of CNTs.

Zone 2: once perfect contact was guaranteed, there was a drop in strain rate and an increase in strength.

Zone 3: an increase in strength became stable and reached the saturation level while the strain rate gradually decreased to zero value. The sample reached maximum compression stress in this zone.

Zone 4: in this zone, the specimen rebounded and started to relax. The strain rate started to decrease below zero value and there was a drop in the stress of the specimen. This situation could be justified by the spring-back behavior of the specimen. At the end of this zone, both signals were negated at approximately the same time.

The damage tolerance is a significant criterion for the composites to be used in the civil and military applications like naval or aerospace. This characteristic goes through the damage behavior of structure from the initial state to final fracture; and numerous methods have been utilized to verify the magnitude of the damage. A high-speed camera was used to monitor and record the behavior of the specimen during the dynamic compression tests performed at 4 bar. The images, which were taken in real-time, show the progression of damage, Figure 13. However, it should be kept in mind that no macroscopic damage in the nanocomposite specimens was noticed at the 4 bar; and the absence of the second peak in the strain rate vs. time curve validated this phenomenon. However, damage at microscopic and nano scale, such as plastic deformation, micro-buckling, kink-bands, and crack could have happened. A damaged zone was observed at 0.12 ms of impact time in the case of 0% CNTs sample, but no damage was obtained in the case of reinforced specimens with CNTs. The incorporation of CNTs not only increase the strength of the material but also played a vital role in delaying the crack propagation phenomena, thus increasing the resistance of material to final fracture.

## 6. Conclusion

An experimental investigation was carried out to study the effect of different wt.% of CNTs on the mechanical energy balance of CFRP nanocomposites using split Hopkinson pressure bars. Samples with different CNT weight percentages (0% as reference, 1% and 2%) were subjected to different incident waves; and results showed that the ability of the material to absorb a mechanical shock wave was improved by increasing the CNTs mass fractions. This increase was due to the enhancement of elastic strain behavior of the composite and decrease in its plasticity with the addition of CNTs in the matrix of the composite. Moreover, damage modes were evaluated, and the results indicated that no macroscopic damage was observed in the specimen under the impact pressures, because CNTs act as a barrier to crack propagation; however, micro cracks and permanent plastic deformation could be present within the nano composites. Thus, the presence of CNTs resulted in greater energy dissipation and increase in energy absorption behavior of these nanocomposites. Results confirmed that for out-of-plane tests, CNT-based nanocomposites exhibited better stiffness and resistance to damage compared to neat material; and dynamic response revealed that the composite final failure was delayed by increasing the CNT % in addition to improvement in energy absorption and dissipation system. Therefore, CNTs might be good nanofillers, which improves the dynamic properties of the composites and enhance the resistance to damage and the energy absorption capability of composite materials for high velocity impact loadings.

## Figures and Tables

**Figure 1 polymers-12-00194-f001:**
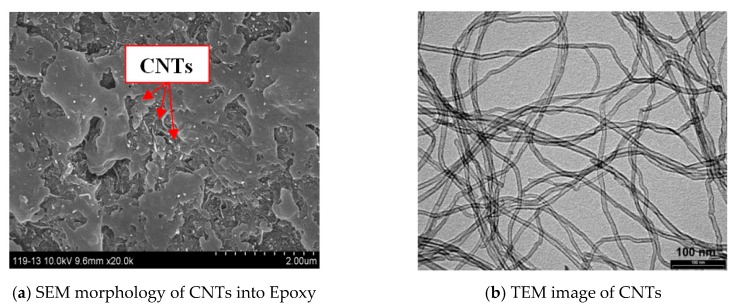
The morphology of multiwall CNTs by (**a**) SEM and (**b**) TEM images.

**Figure 2 polymers-12-00194-f002:**
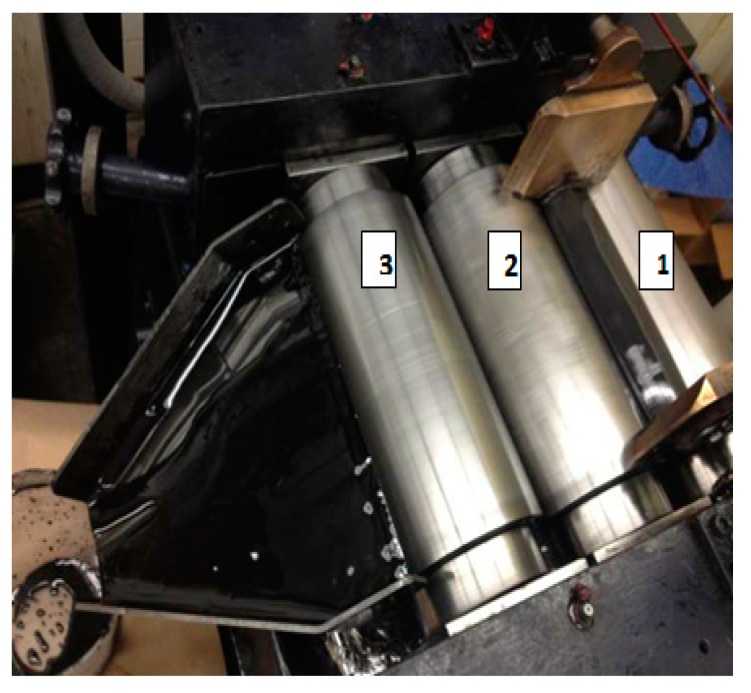
Lehmann Mills three-roll mixer.

**Figure 3 polymers-12-00194-f003:**
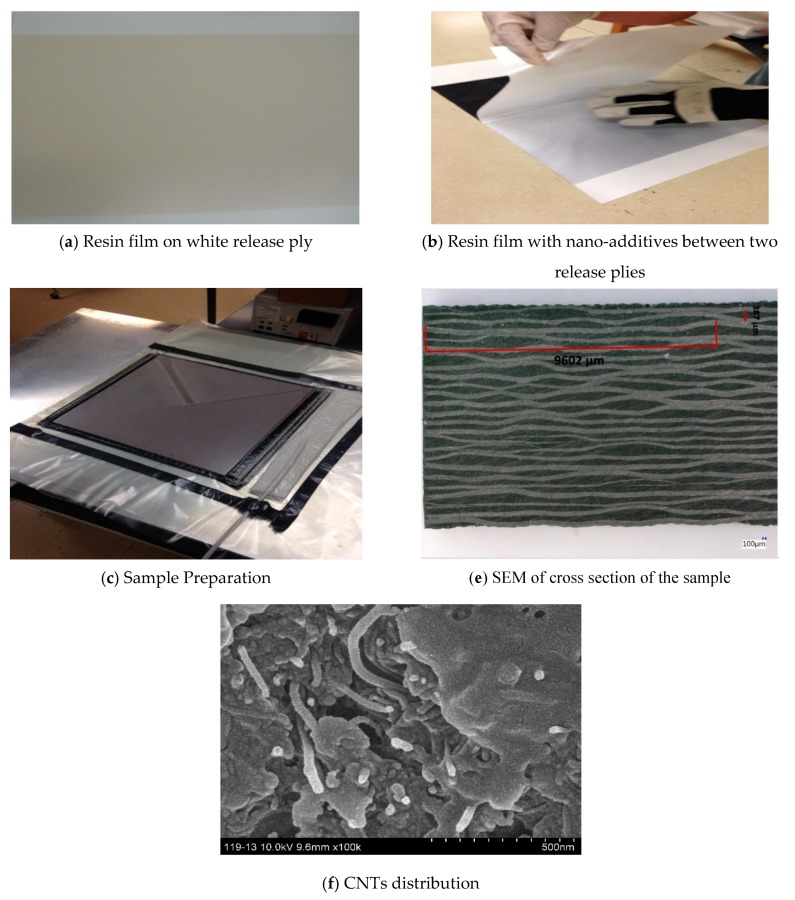
Manufacturing steps.

**Figure 4 polymers-12-00194-f004:**
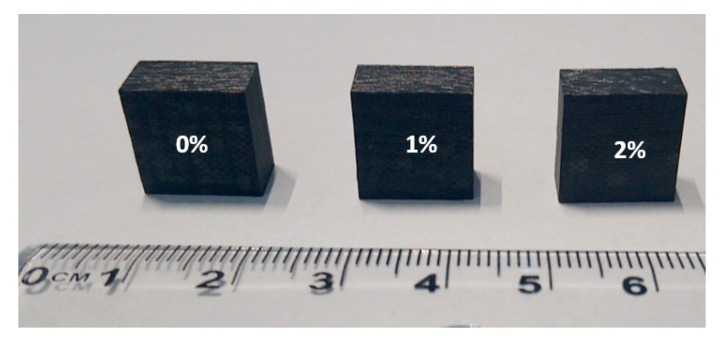
Specimens with different percentages dedicated to dynamic compression tests.

**Figure 5 polymers-12-00194-f005:**
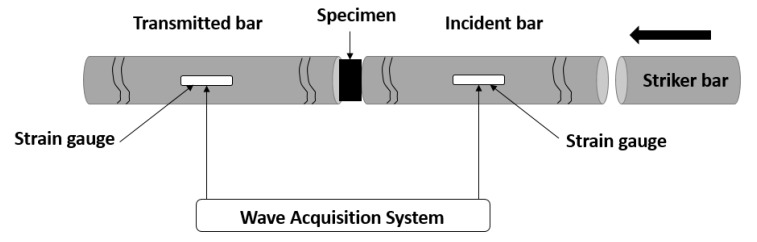
Split Hopkinson Pressure Bar.

**Figure 6 polymers-12-00194-f006:**
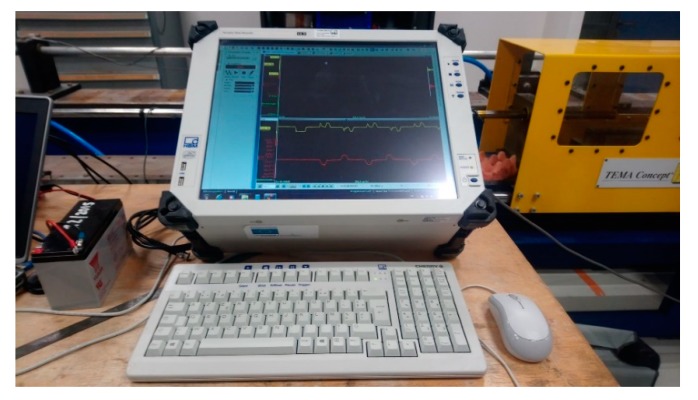
Digital oscilloscope.

**Figure 7 polymers-12-00194-f007:**
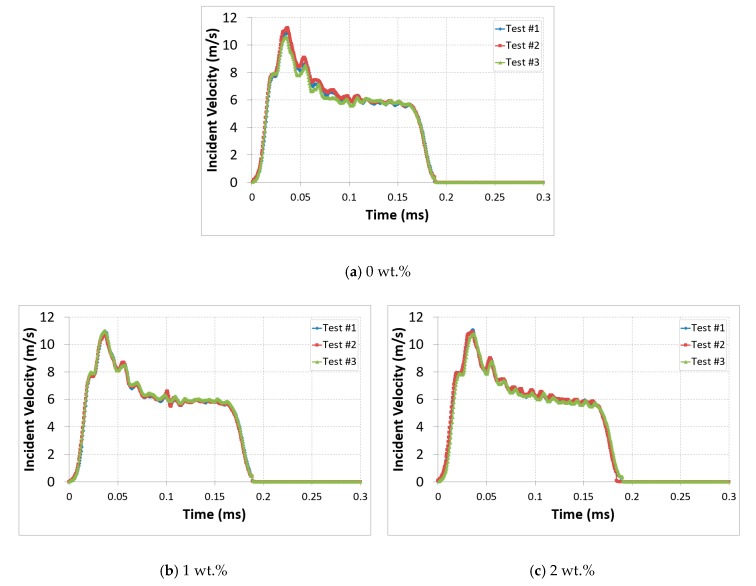
Test reproducibility of the nanocomposites with different CNTs mass fractions, P = 4 bar.

**Figure 8 polymers-12-00194-f008:**
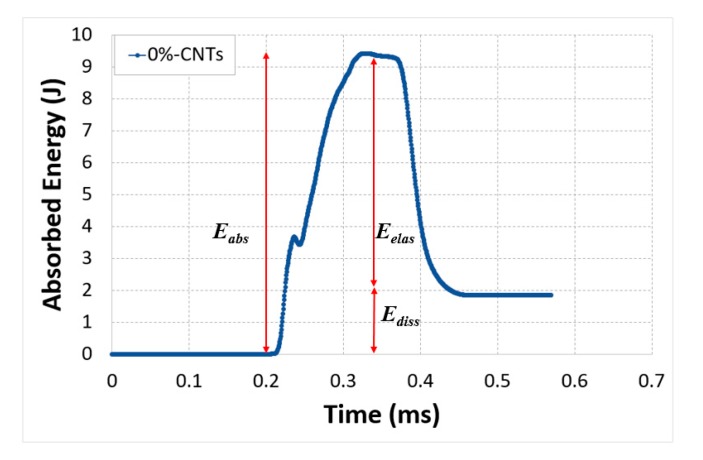
Typical profile of the energy absorption of the tested specimen.

**Figure 9 polymers-12-00194-f009:**
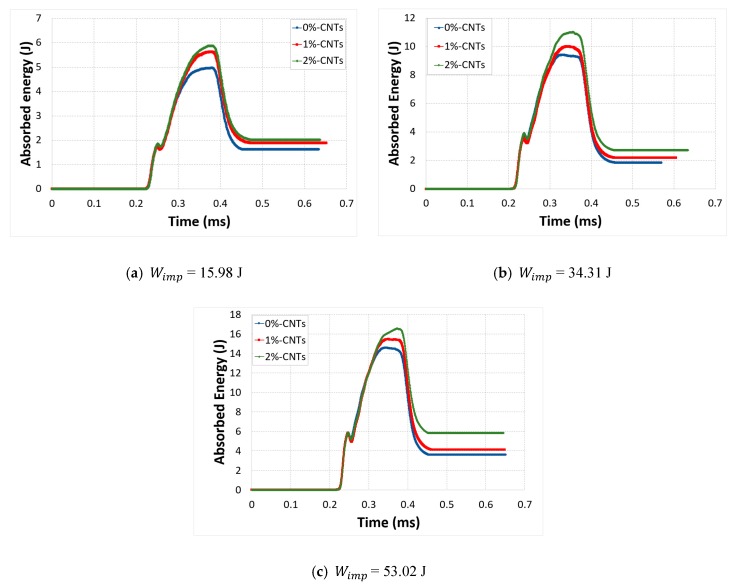
Absorbed energy vs. time for different CNTs mass fractions.

**Figure 10 polymers-12-00194-f010:**
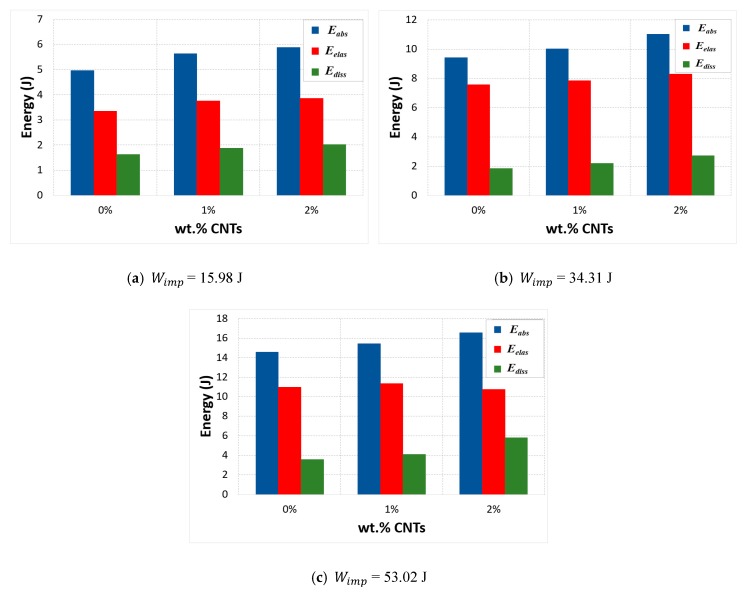
Energy balance vs. CNT mass fraction at different impact energies.

**Figure 11 polymers-12-00194-f011:**
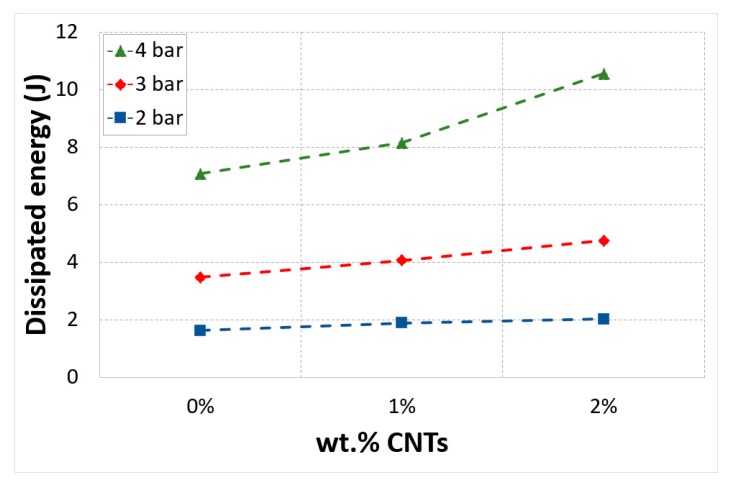
Dissipated energy vs CNTs mass fractions.

**Figure 12 polymers-12-00194-f012:**
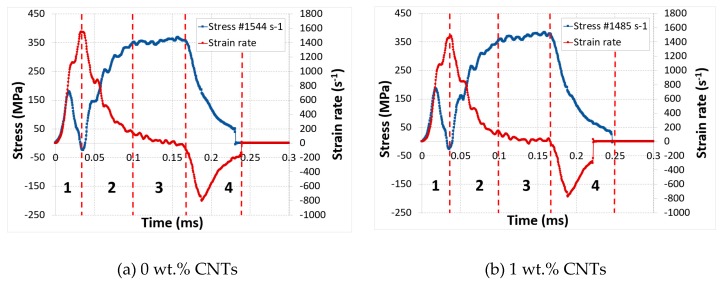
Evolution of the stress and strain rate vs time curves for different mass fraction, P = 4 bar.

**Figure 13 polymers-12-00194-f013:**
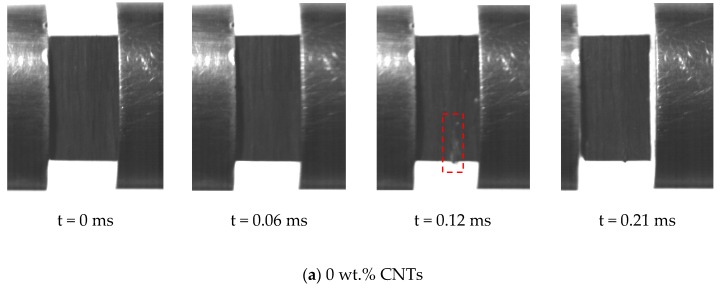
High-speed photograph of real-time dynamic compression test of nanocomposites with different CNTs mass fraction, P = 4 bar.

**Table 1 polymers-12-00194-t001:** Material properties.

Carbon fiber	Epoxy matrix	CNT
E_11_ (GPa)	230	E (GPa)	2.72	E (GPa)	500
E_22_ (GPa)	15	v	0.3	v	0.261
E_33_ (GPa)	15				
v_12_	0.28				
v_13_	0.28				
v_23_	0.28				
G_12_ (GPa)	15				
G_13_ (GPa)	15				
G_23_ (GPa)	15

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
