# Peer review of "Effect of CNTs Additives on the Energy Balance of Carbon/Epoxy Nanocomposites during Dynamic Compression Test"

_polymers, 2020, doi:10.3390/polym12010194_

Round 1

Reviewer 1 Report

The manuscript is written in understable and clear way. However, some changes should be made. 

Why do  you write Carbon nanotubes? I think that carbon nanotubes is better, there is no need to write it with capital letter C. As well as write, please, Split Hopkinson Pressure Bar or split Hopkinson pressure bar devices (in abstract).

Line 57: I do not agree that CNTs are members of fullerene family. Better is carbon nanomaterials family or similar.    Line 91-92: describe, please, the method of CNTs production. Line 96-98: could you precise a type of laboratory mixer and three‐roller paint mill? Format of table 1 should be corrected, it should be aligned. Line 135-138: This part should be involved in previous part 3. Test procedure. As well as part in lines 142-176 is a theoretical part and should be implemented in part 3.

How does the text correspond: line 179-180 .... energy absorption behavior of each specimen was studied at impact energy of 15 180 J, 34 J and 53 J ... to the caption of the Figure 7 ...Absorbed energy vs. time for different CNTs mass fractions: (a) 2 bar; (b) 3bar; (c) 4 bar... ? I think it should be unified. 

In Figure 7 c: Axis x should be corrected.

In Figure 8: In caption you should write Wimp like in the text, indexed and in Italic style. You should explain W_elas, W_ abs, W_diss. It is not in the text explained using these short names.

Figure 9: Caption should start with a capital letter, also improve, please, format of the description of axes - unify it with other figures.

Line 225: Why did you use P=4bars? Explain, please.

Line 230: No capital letter at the beginning.

Line 231: Stress or strength? 

Author Response

Why do you write Carbon nanotubes? I think that carbon nanotubes is better, there is no need to write it with capital letter C. As well as write, please, Split Hopkinson Pressure Bar or split Hopkinson pressure bar devices (in abstract)

The modifications has been made as requested. Carbon nanotubes is replaced by carbon nanotubes and Split Hopkinson Pressure bar device is replaced by Split Hopkinson Pressure Bar (Please see line 13, line 18).

Line 57: I do not agree that CNTs are members of fullerene family. Better is carbon nanomaterials family or similar.    Line 91-92: describe, please, the method of CNTs production. Line 96-98: could you precise a type of laboratory mixer and three‐roller paint mill? Format of table 1 should be corrected, it should be aligned. Line 135-138: This part should be involved in previous part 3. Test procedure. As well as part in lines 142-176 is a theoretical part and should be implemented in part 3

Fullerene family is replaced by carbon nanomaterials family (Please see line 58) as requested Multi-walled carbon nanotubes (MWNTCs) were provided by nanocyl Belgium Company, it has been added in the text (Please see line 122). An example of the morphology of this materials obtained by SEM and AFM has also been added in the manuscript as Figure 1. The type of laboratory mixer and three-roller paint mill has been mentioned as requested (please see line 141 and line 143) The format of table 1 has been updated as requested Line 135-138: this part has been added in part 3 as requested (Please see line 177-206) Line 142-176 this part has been implemented in part 3 as requested (Please see line208-243)

How does the text correspond: line 179-180.... energy absorption behavior of each specimen was studied at impact energy of 15 180 J, 34 J and 53 J ... to the caption of the Figure 7 ...Absorbed energy vs. time for different CNTs mass fractions: (a) 2 bar; (b) 3bar; (c) 4 bar... ? I think it should be unified.

- line 179-180 and the caption of the figure 7 has been unified as requested (Please see line 247-248 and Figure 9)

In Figure 7 c: Axis x should be corrected.

- Axis x in Figure 7 is corrected as requested (Please see Figure 9)

In Figure 8: In caption, you should write Wimp like in the text, indexed and in Italic style. You should explain W_elas, W_ abs, W_diss. It is not in the text explained using these short names.

- In figure 8 Wimp is modified like in the text, indexed and in italic style as requested (Please see Figure 10)

- In figure 8 W_elas, W_ abs, W_diss in the legend are replaced by    and  as mentioned in the text. They have the same explanation as W_elas, W_ abs, W_diss (Please see Figure 10)

Figure 9: Caption should start with a capital letter, also improve, please, format of the description of axes - unify it with other figures.

- Caption of Figure 9 has been updated as requested, it starts with a capital letter and description of the axes has been improved (Please see Figure 11)

Line 225: Why did you use P=4bars? Explain, please.

4 bar is the maximum pressure that can be achieved in the SPHB machine so 4 bar was our limit. We have performed tests at different pressures ranging from 2 to 4 bars but 4 bar was the pressure showing damage in samples (0%). We wanted to see the effect of CNTs on the improvement of damage mechanism of fiber reinforced composites so we chose 4bar to present the diversity of the behavior of our material (Please see line178-185).

Line 230: No capital letter at the beginning.

- The capital letter is added as requested ( please see line 304).

Line 231: Stress or strength? 

- Stress was replaced by Strength (Please see 305).

Reviewer 2 Report

The authors studied the effects of CNTs on mechanical energy balance of CFRP nanocomposites. Increased energy absorption capability was found with increasing CNTs, which is a complement for previous CNTs/CFRP nanocomposite studies. However, only samples with two CNTs mass fraction were studied. Would the addition of CNTs increase the energy absorption capability with other mass fraction? The authors may detail their investigations in their following works.

Author Response

The authors studied the effects of CNTs on mechanical energy balance of CFRP nanocomposites. Increased energy absorption capability was found with increasing CNTs, which is a complement for previous CNTs/CFRP nanocomposite studies. However, only samples with two CNTs mass fraction were studied. Would the addition of CNTs increase the energy absorption capability with other mass fraction? The authors may detail their investigations in their following works.

In this research paper, the results have shown an improvement of dynamic behavior of fiber reinforced composites by the integration of CNTs. This study was conducted because authors wanted to understand whether CNTs have any effect on the behavior of composites and what happens if you further increase the wt.% of these nano fillers. The results confirmed that the addition of CNTs even with small percentage improves the damage characteristics of these materials under dynamic loading and did not act as inclusion or defect. 

However, in our next stage of research, we are currently working on adding increased wt% of CNTs in epoxy in fiber reinforced material and find the percolation point and also find optimum wt% of CNTs to achieve best dynamic response. Then we will study the best compromise between the cost, good performance and weight of the structure. The remark of the reviewer will benefit us in our future research which is solely dedicated to understand the effect of higher mass fractions of CNTs and find a best compromise between cost quality and weight of the structure.

Reviewer 3 Report

The study is of some interest with good experimental work but I consider the paper below the standard required for publication in your journal. The introduction is very general in nature. There is a general lack of detail in the experimental section (for example: No sufficient details are given for the epoxy matrix and CNTs (presumably this is made to a specification?) Supplier of these materials?. Also, it is not clear how the CNTs/Epoxy films were fabricated. The range of CNTs % used in this study is very narrow and doesn't give a clear image of the pattern this material can take with lower and higher loadings. Microscopy investigation would be good to be considered to examine the dispersion and bonding of CNTs to the polymer and relate it to the results obtained. The results should have been analysed in a better way to clearly show the differences between the CNTs loadings and the impact energies used. There is a little real discussion of the results and the conclusion which does not convey to the reader the significance or novelty of the findings.

Anyway, I suggest the manuscript to be accepted. But the authors should improve on the whole manuscript and particularly the analysis and discussion of the results. The final decision of the editor is paramount.

Author Response

The study is of some interest with good experimental work but I consider the paper below the standard required for publication in your journal. The introduction is very general in nature. There is a general lack of detail in the experimental section (for example: No sufficient details are given for the epoxy matrix and CNTs (presumably this is made to a specification?) Supplier of these materials?. Also, it is not clear how the CNTs/Epoxy films were fabricated. The range of CNTs % used in this study is very narrow and doesn't give a clear image of the pattern this material can take with lower and higher loadings. Microscopy investigation would be good to be considered to examine the dispersion and bonding of CNTs to the polymer and relate it to the results obtained. The results should have been analysed in a better way to clearly show the differences between the CNTs loadings and the impact energies used. There is a little real discussion of the results and the conclusion which does not convey to the reader the significance or novelty of the findings.

- The technical language and grammar of the paper has been improved as requested.

- Introduction has been modified as requested.

- Experimental section is improved (the information about the types and the suppliers of epoxy, carbon fiber and CNTs used in this investigation are now mentioned in the paper as requested (Please see section 2)

- Manufacturing process is also mentioned in the section 2

- An example of the morphology obtained by SEM and AFM is also added in the manuscript and shown in Figure 1. Mechanical and morphological properties are listed in section 2. 

- Results are further elaborated in detail as requested.  

- Conclusion is further improved as requested.

Round 2

Reviewer 1 Report

Dear Authors, 

the changes are well done. 

Please, before publishing, correct:

citation [27] in Introduction, line 70. The author's name is Prachař - maybe is better write Prachar if it is not possible to write it correctly - use symbol in Word. The same correction must be done in References.

Part 2, line 121-122: it should be written ....and multi‐walled carbon nanotubes (MWNTCs) were 122 produced by Nanocyl Belgium Company,

Author Response

From Dr. Manel CHIHI to

Reviewers of Polymers Journal.

Reply to the revision of the paper N°: polymers-655491

Effect of CNTs Additives on the Energy balance of carbon/epoxy nanocomposites during dynamic compression test

By: M. Chihi, M. Tarfaoui, C. Bouraoui and A. El Moumen

Dear colleagues,
Thank you for reviewing our paper and the time invested in that endeavor. We have reorganized and numbered your comments so as to make sure we address all the raised issues. The text below lists the reviewer’s comments (in italics) and our replies, for completeness. All the remarks provided below refer to the revised version attached.
We have made corrections according the reviewer comments. All these modifications are highlighted in the revised document in red color 

Comments from the editors and reviewers:

Reviewer 1

citation [27] in Introduction, line 70. The author's name is Prachař - maybe is better write Prachar if it is not possible to write it correctly - use symbol in Word. The same correction must be done in References.

The author’s name was corrected (Please see line 69 , line 407)

Part 2, line 121-122: it should be written ....and multi‐walled carbon nanotubes (MWNTCs) were 122 produced by Nanocyl Belgium Company,

The modifications have been made as requested. Multi‐walled carbon nanotubes (MWNTCs) was replaced by multi‐walled carbon nanotubes (MWNTCs) and nanocyl Belgium Company was replaced by Nanocyl Belgium Company (Please see line 120, line 121)
